# SimPLR: A Simple and Plain Transformer for Object Detection and Segmentation

## Abstract

The ability to detect objects in images at varying scales has played a pivotal role in the design of modern object detectors. Despite considerable progress in removing handcrafted components using transformers, multi-scale feature maps remain a key factor for their empirical success, even with a plain backbone like the Vision Transformer (ViT). In this paper, we show that this reliance on feature pyramids is unnecessary and a transformer-based detector with scale-aware attention enables the plain detector 'SimPLR' whose backbone *and* detection head both operate on single-scale features. The plain architecture allows SimPLR to effectively take advantages of self-supervised learning and scaling approaches with ViTs, yielding strong performance compared to multi-scale counterparts. We demonstrate through our experiments that when scaling to larger backbones, SimPLR indicates better performance than end-to-end detectors (Mask2Former) and plain-backbone detectors (ViTDet), while consistently being faster. The code will be released.

## 1 Introduction

After its astonishing achievements in natural language processing, the transformer (Vaswani et al., 2017) has quickly become the neural network architecture of choice in computer vision, as evidenced by recent success in image classification (Liu et al., 2021; Dosovitskiy et al., 2021), object detection (Carion et al., 2020; Zhu et al., 2021; Nguyen et al., 2022) and segmentation (Wang et al., 2021a; Zhang et al., 2021; Cheng et al., 2022). Unlike natural language processing, where the same pre-trained network can be deployed for a wide range of downstream tasks with only minor modifications (Brown et al., 2020; Devlin et al., 2019), computer vision tasks such as object detection and segmentation require a different set of domain-specific knowledge to be incorporated into the network. Consequently, it is commonly accepted that a modern object detector contains two main components: a pre-trained backbone as the *general* feature extractor, and a *task-specific* head that conducts detection and segmentation tasks using domain knowledge. For transformer-based vision architectures, the question remains whether to add more inductive biases or to learn them from data.

The spatial nature of image data lies at the core of computer vision. Besides learning long range feature dependencies, the ability of capturing local structure and relationships between neighboring pixels is critical for representing and understanding the image content. Building upon the successes of convolutional neural networks in computer vision, a line of research biases the transformer architecture to be *multi-scale* and *hierarchical* when dealing with the image input, *i.e.*, Swin Transformer by Liu et al. (2021) and others (Fan et al., 2021; Wang et al., 2021b; Heo et al., 2021). The hierarchical design makes it easy to create multi-scale features for dense vision tasks and allows pre-trained transformers to be seamlessly integrated into convolution-based detection head with feature pyramid network (Lin et al., 2017), yielding impressive results in object detection and segmentation. However, the inductive biases in the architectural design make it benefit less from the self-supervised learning and the scaling of model size (Li et al., 2022).

An alternative direction pursues the idea of a simple transformer with "less inductive biases" and to emphasize learning vision-specific knowledge directly from image data. Specifically, the Vision Transformer (ViT) (Dosovitskiy et al., 2021) stands out as a *plain* and *non-hierarchical* architecture with a constant feature resolution, and acts as the feature extractor in plain-backbone detection. This is motivated by the success of ViTs scaling behaviours in visual recognition (He et al., 2022; Bao et al., 2022; Dehghani et al., 2023). In addition, the end-to-end detection framework proposed by Carion et al. (2020) with a transformer-based detection head further removes many hand-designed

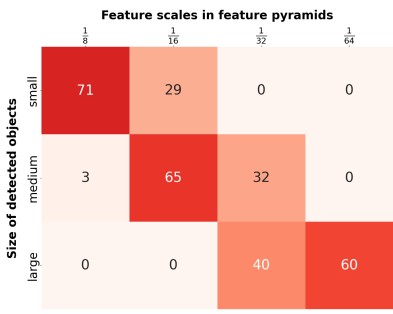
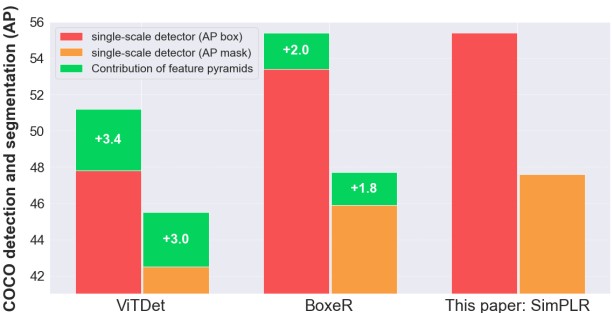

(a) **Strong correlation** between feature scales in feature pyramids and object sizes.

(b) **Feature pyramids are important**, showing a considerable improvement over single-scale input in current detectors.

Figure 1: **A single-scale detector is non-trivial.** **(a)** We visualize the alignment between feature scales in the feature pyramids and the size of COCO detected objects in BoxeR (Nguyen et al., 2022). Here, we adopt SimpleFPN with ViT from (Li et al., 2022) as backbone for BoxeR. A strong correlation between feature scale and size of detected objects suggests that feature pyramids remove the difficulties in detecting objects of various sizes. **(b)** Even with a pre-trained ViT backbone, feature pyramids are important for both convolution-based (ViTDet) and transformer-based (BoxeR) detectors. In this paper, however, we demonstrate that the plain detector[1], SimPLR, yields competitive performance compared to multi-scale counterparts, and the scale-aware attention in its transformer-based detection head is able to capture multi-scale patterns from single-scale input.

components, like non-maximum suppression or intersection-over-union computation, that encodes the prior knowledge for object detection.

The plain design of ViTs, however, casts doubts about its ability to capture information of objects across multiple scales. While recent studies (Dosovitskiy et al., 2021; Li et al., 2022) suggest that ViTs with global self-attention could potentially learn translation-equivariant and scale-equivariant properties during training, leading object detectors still require multi-scale feature maps. This observation holds true for both convolutional (Ren et al., 2015; He et al., 2017) and transformer-based detectors (Zhu et al., 2021; Nguyen et al., 2022; Cheng et al., 2022) (see Fig. 1). Unlike hierarchical backbones, the creation of feature pyramids conflicts with the original design philosophy of ViTs. We postulate that a scale-aware attention mechanism would enable the transformer-based detector to predict objects of various sizes using only a single-scale feature map, thus eliminating the need to accommodate complex modifications for feature pyramids and enabling the network to acquire scale information from data.

In this paper, we introduce SimPLR, a simple detector where both the backbone *and* the detection head operate on plain features. Our network architecture follows the end-to-end framework. In particular, SimPLR extracts the single-scale feature map from a pre-trained ViT, which is then fed into the transformer encoder-decoder via a simple projection to make the prediction. In order to detect objects of various sizes, we propose to learn scale information directly in the attention computation of the encoder, resulting in an *adaptive-scale* attention mechanism. This eliminates the need for multi-scale feature maps, yielding a simple and efficient detector. With minimal adaptations for panoptic segmentation, we show the effectiveness of the proposed single-scale detector, SimPLR, on three tasks of the COCO dataset: object detection, instance segmentation, and panoptic segmentation. Notably, the efficient design allows SimPLR to take advantages of the significant progress in self-supervised learning and scaling ViTs, indicating compelling scaling behaviour with larger models.

## 2 BACKGROUND

Our goal is to further simplify the object detection pipeline from Li et al. (2022); Nguyen et al. (2022), and to prove the effectiveness of the plain detector in object detection and segmentation tasks. To do so, we focus on the recent progress in end-to-end object detection. Specifically, we utilize the

---

[1]In this paper, "backbone" refers to the components that we inherit from the pre-training stage, "detection head" refers to the components that are initialized from scratch, and "plain" refers to the single-scale property. The "plain detector" is the detector whose backbone and head both operate on single-scale features.

box-attention mechanism by Nguyen et al. (2022) as a strong baseline due to its effectiveness in learning discriminative object representations while being lightweight in computation.

Given the input feature map from the backbone, the encoder layers with box-attention will output contextual representations. The contextual representations are utilized to predict object proposals and to initialize object queries for the decoder. We denote the input feature map of an encoder layer as $e \in \mathbb{R}^{H_e \times W_e \times d}$ and the query vector $q \in \mathbb{R}^d$, with $H_e, W_e, d$ denoting height, width, and dimension of the input features respectively. Each query vector $q \in \mathbb{R}^d$ in the input feature map is assigned a reference window $r=[x, y, w, h]$, where $x, y$ indicate the query coordinate and $w, h$ are the size of the reference window both being normalized by the image size. The box-attention refines the reference window into a region of interest, $r'$, as:

$$r' = F_{\text{scale}}\big(F_{\text{translate}}(r, q), q\big), \tag{1}$$

$$F_{\text{scale}}(r, q) = [x, y, w + \Delta_w, h + \Delta_h], \qquad F_{\text{translate}}(r, q) = [x + \Delta_x, y + \Delta_y, w, h], \tag{2}$$

where $F_{\text{scale}}$ and $F_{\text{translate}}$ are the scaling and translation transformations, $\Delta_x, \Delta_y, \Delta_w$ and $\Delta_h$ are the offsets regarding to the reference window $r$. A linear projection ($\mathbb{R}^d \to \mathbb{R}^4$) is applied on $q$ to predict offset parameters (*i.e.*, $\Delta_x, \Delta_y, \Delta_w$ and $\Delta_h$) w.r.t. the window size.

Similar to self-attention (Vaswani et al., 2017), box-attention aggregates $n$ multi-head features from regions of interest:

$$\text{MultiHeadAttention} = \text{Concat}(\text{head}_1, \ldots, \text{head}_n) W^O. \tag{3}$$

During the $i$-th attention head computation, a $2 \times 2$ feature grid is sampled from the corresponding region of interest $r'_i$, resulting in a set of value features $v_i \in \mathbb{R}^{2 \times 2 \times d_h}$. The $2 \times 2$ attention scores are efficiently generated by computing a dot-product between $q \in \mathbb{R}^d$ and relative position embeddings ($k_i \in \mathbb{R}^{2 \times 2 \times d}$) followed by a softmax function. The attended feature $\text{head}_i \in \mathbb{R}^{d_h}$ is a weighted average of the $2 \times 2$ value features in $v_i$ with the corresponding attention weights:

$$\alpha = \text{softmax}(q^\top k_i), \qquad \text{head}_i = \text{BoxAttention}(q, k_i, v_i) = \sum_{j=0}^{2 \times 2} \alpha_j v_{i_j}, \tag{4}$$

where $q \in \mathbb{R}^d$, $k_i \in \mathbb{R}^{2 \times 2 \times d}$, $v_i \in \mathbb{R}^{2 \times 2 \times d_h}$ are query, key and value vectors of box-attention, $\alpha_j$ is the $j$-th attention weight, and $v_{i_j}$ is the $j$-th feature vector in the feature grid $v_i$. To better capture objects at different scales, the box-attention (Nguyen et al., 2022) takes $t$ multi-scale feature maps, $\{e^j\}_{j=1}^t$, as its inputs. In the $i$-th attention, $t$ feature grids are sampled from each of multi-scale feature maps in order to produce $\text{head}_i$.

By computing the attended feature within regions of interest in each attention head, box-attention shows strong performance in object detection and instance segmentation with a small computational budget. The transformation functions (*i.e.*, translation and scaling) allow box-attention to capture long-range dependencies. The effectiveness and efficiency of box-attention mechanism brings up the question: *Is multi-scale object information learnable within a single-scale feature map?*

## 3 SimPLR: A Simple and Plain Single-Scale Object Detector

Multi-scale feature maps in a hierarchical backbone can be easily extracted from the pyramid structure (Liu et al., 2016; Lin et al., 2017; Zhu et al., 2021). When moving to a ViT backbone with a constant feature resolution, the creation of multi-scale feature maps requires complex backbone adaptations. Moreover, the benefits of multi-scale features in object detection frameworks using ViTs remain unclear. Recent studies on plain-backbone detection (Li et al., 2022; Chen et al., 2022) conjecture the high-dimensional ViT with self-attention and positional embeddings (Vaswani et al., 2017) is able to preserve important information for localizing objects[2]. From this conjecture, we hypothesize that a proper design of the transformer-based head will enable a single-scale detector.

Our proposed plain detector, SimPLR, is conceptually simple: a pre-trained ViT backbone to extract plain features from an image, which are then fed into a single-scale encoder-decoder to make the final prediction (See Fig. 2). Thus, SimPLR is a natural idea as it eliminates the non-trivial creation of feature pyramids from the ViT backbone. But the single-scale encoder-decoder requires an effective design to deal with objects at different scales. Next, we introduce the key elements of SimPLR, including its *scale-aware* attention that is the main factor for learning of adaptive object scales.

---

[2]ViT-B and larger (dim $\geq$ 768) can maintain information with a patch size of $16 \times 16 \times 3$ in the input image.

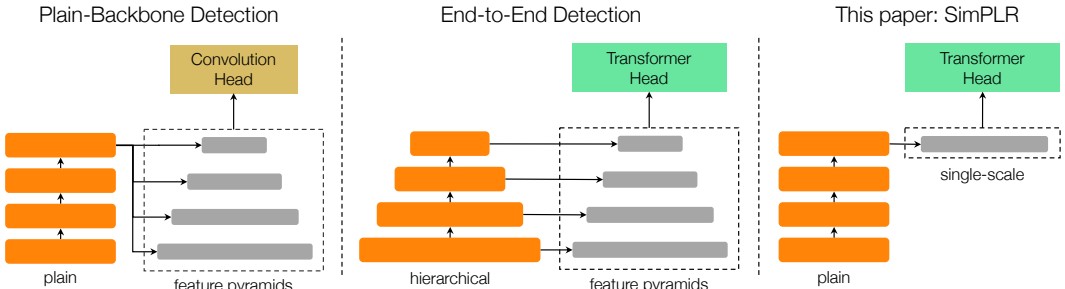

Figure 2: **Object detection architectures. Left:** The plain-backbone detector from Li et al. (2022) whose input (denoted in the dashed region) are multi-scale features. **Middle:** State-of-the-art end-to-end detectors (Nguyen et al., 2022; Cheng et al., 2022) utilize a hierarchical backbone (*i.e.*, Swin (Liu et al., 2021)) to create multi-scale inputs. **Right:** Our simple single-scale detector following the end-to-end framework. Where existing detectors require feature pyramids to be effective, we propose a plain detector, SimPLR, whose backbone and detection head operate on a single-scale feature map. Using only single-scale input, SimPLR achieves on par or even better performance compared to multi-scale counterparts while being more efficient.

**Single-scale detector.** SimPLR follows the end-to-end detection framework proposed by Carion et al. (2020) with the two-stage design. Specifically, we use a plain ViT as the backbone with $14 \times 14$ windowed attention and four equally-distributed global attention blocks as in Li et al. (2022). In the detection head, the SimPLR encoder receives *single-scale* features via a projection of the last feature map from the ViT backbone. The object proposals are then predicted by the encoder and top-scored features from the plain feature map are initialized as object queries for the SimPLR decoder to predict bounding boxes and masks. Note this design differs from most state-of-the-art systems, where the feature pyramids are essential for object proposal generation and final prediction. Our approach follows the spirit of ViT (Dosovitskiy et al., 2021) that applies a single and constant feature resolution throughout the detection head, which turns out to largely simplify the architectural choices and remove handcrafted components compared to Swin (Liu et al., 2021) or MViT (Fan et al., 2021).

Formally, we apply a projection $f$ to the last feature map of the pre-trained ViT backbone, resulting in the input feature map $e \in \mathbb{R}^{H_e \times W_e \times d}$ where $H_e, W_e$ are the size of the feature map, and $d$ is the hidden dimension of the detection head. In SimPLR, the projection $f$ is simply a single convolution projection, that provides us a great flexibility to control the resolution and dimension of the input features $e$ to the encoder. The projection allows SimPLR to decouple the feature scale and dimension between its backbone and detection head. To be specific, the ViT backbone with global self-attention can operate efficiently on high-dimensional and low-resolution features, while the transformer-based detection head with sparse attention takes the lower dimension but higher resolution input for better object prediction. This practice is different from the creation of SimpleFPN in Li et al. (2022) where a different stack of multiple convolution layers is used for each feature scale (more details shown in Fig. A in the supplementary material). We show by experiments that this formulation is key for single-scale detection and segmentation while keeping our network efficient.

**Scale-aware attention.** The output features of the encoder should capture objects at different scales. Therefore, unlike the feature pyramids where each set of features encode a specific scale, predicting objects from a plain feature map requires its feature vectors to reason about dynamic scale information based on the image content. This can be addressed effectively by a multi-head attention mechanism that capture different scale information in each of its attention heads. In that case, global self-attention is a potential candidate because of its large receptive field and powerful representation. However, its computational complexity is quadratic w.r.t. the sequence length of the input, making the operation computationally expensive when dealing with high-resolution images. The self-attention also leads to worse performance and slow convergence in end-to-end detection (Zhu et al., 2021). This motivated us to develop a multi-head *scale-aware* attention mechanisms based on the box-attention.

The multi-head attention mechanism, proposed by Vaswani et al. (2017), is a core operation in the transformer architectures for capturing diverse patterns given a query vector in the input features. In the multi-head box-attention (Nguyen et al., 2022), each feature vector is assigned a reference window which is then refined to locate a region-of-interest in each attention head via scaling and translation transformations. Theoretically, the scaling transformation should provide box-attention the

capability to learn multi-scale regions-of-interest in multiple attention heads. However, we find in our experiments that the scaling function in box-attention only performs minor modifications regarding to its reference window. As a result, feature vectors learn to adapt to a specific scale of the reference window assigned to them. While this behaviour may not impact the multi-scale box-attention – which utilizes feature pyramids for detecting objects – it poses a big challenge in learning scale-equivariant features on a single-scale input.

To address this limitation, we propose two variants of multi-head *scale-aware* attention (*i.e.*, *fixed-scale* and *adaptive-scale*) that integrate different scales into each attention head, allowing query vectors to choose the suitable scale information during training. Our proposed attention mechanism is simple: we assign reference windows of $m$ different scales to attention heads of each query. We use $m$ reference windows with size $w=h \in \left\{ s \cdot 2^j \right\}_{j=0}^{m-1}$, where $s$ is the size of the smallest window, and $m$ is the number of scales with the growing rate of 2. Surprisingly, our experiments show that the results are not sensitive to the size of the window, as long as *enough* number of scales are used.

i) *Fixed-Scale Attention.* Given reference windows of $m$ scales, we distribute them to $n$ attention heads in a round-robin manner. Thus, in multi-head fixed-scale attention, $\frac{n}{m}$ attention heads are allocated for each of the window scales. This uniform distribution of different scales enables fixed-scale attention to learn diverse information from local to more global context. The aggregation of $n$ heads results in scale-aware features, that is suitable for predicting objects of different sizes.

ii) *Adaptive-Scale Attention.* Instead of uniformly assigning $m$ scales to $n$ attention heads, the adaptive-scale attention learns to allocate a scale distribution based on the context of the query vector. This comes from the motivation that the query vector belonging to a small object should use more attention heads for capturing fine-grained details rather than global context, and vice versa. More specifically, in each attention head, the adaptive-scale attention predicts offsets for all reference windows of $m$ scales and samples feature grids from $m$ regions-of-interest. Given the query vector $q \in \mathbb{R}^d$, it then applies a learnable projection on $q$ followed by $\mathrm{softmax}$ normalization to generate attention scores which allow it to focus on the feature grid of suitable scale. The adaptive-scale attention provides efficiency due to sparse sampling and strong flexibility to control scale distribution via its attention computation.

**Plain backbone.** SimPLR deploys ViT as its plain backbone for feature extraction. We show that SimPLR can take advantages of recent progress in self-supervised learning with ViTs. To be specific, SimPLR generalizes to ViT backbones initialized by MAE (He et al., 2022) and BEiTv2 (Peng et al., 2022). The efficient design of SimPLR allows us to effectively scale to larger ViT backbones which recently show to be even more powerful in learning representations (He et al., 2022; Zhai et al., 2022; Dehghani et al., 2023). We also provide the comparison between different pre-training approaches of the SimPLR backbone in the supplementary material.

**Adaption for panoptic segmentation.** Panoptic segmentation proposed by Kirillov et al. (2019) requires the network to segment both "thing" and "stuff". To enable the plain detector on panoptic segmentation, we make an adaptation in the mask prediction of SimPLR. To be specific, we predict segmentation masks of both types by computing the dot-product between object queries and a feature map following (Cheng et al., 2022). We provide a brief description on these modifications, the full implementation details are provided in the supplementary material.

In Cheng et al. (2022), the $\frac{1}{4}$ feature scale is extracted from the first stage of the Swin and combined with upscaled $\frac{1}{8}$ features from the last layer of the encoder for the mask prediction. As the ViT and SimPLR encoder features are of lower resolution, we simply interpolate the last encoder layer to $\frac{1}{4}$ scale and add a single attention layer on top. This simple modification produces a high resolution feature map that is beneficial for learning fine-grained details. Masked instance-attention follows the dense grid sampling strategy (*e.g.*, $14 \times 14$ feature grid) of box-attention in the decoder (Nguyen et al., 2022), but differs in the computation of the attention scores to better capture objects of different shapes. Inspired by masked self-attention (Cheng et al., 2022), we employ the masking to $14 \times 14$ attention scores of the feature grid based on the mask prediction scores in the previous decoder layer. By focusing better on foreground features, the decoder yields more discriminative features.

## 4 EXPERIMENTS

**Experimental setup.** In this study, we evaluate our method on COCO (Lin et al., 2014), a commonly used dataset for object detection, instance segmentation, and panoptic segmentation tasks. By default,

we use a single-scale feature map with adaptive-scale attention as described in Sec. 3; and initialize the backbone from MAE (He et al., 2022) pre-trained on ImageNet-1K without any labels. In both fixed-scale and adaptive-scale attention, we assign 4-scale reference windows of $\{32^2, 64^2, 128^2, 256^2\}$ to their attention heads. Unless specified, the hyper-parameters are the same as in Nguyen et al. (2022). For all experiments, our optimizer is AdamW (Loshchilov & Hutter, 2019) with a learning rate of 0.0001. The learning rate is linearly warmed up for the first 250 iterations and decayed at 0.9 and 0.95 fractions of the total number of training steps by a factor 10. ViT-B (Dosovitskiy et al., 2021) is set as the backbone. The input image size is $1024 \times 1024$ with large-scale jitter (Ghiasi et al., 2021) between a scale range of $[0.1, 2.0]$. Due to the limit of our computational resources, we report the ablation study using the standard $5\times$ schedule setting with a batch size of 16 as in Nguyen et al. (2022). In the main experiments, we use the finetuning recipe from Li et al. (2022). The same settings are applied to all three tasks.

**A single-scale detector is non-trivial.** In Fig. 1, we explore the use of single-scale input feature by simply projecting the last feature map ($\frac{1}{16}$ scale) of the ViT backbone to the detection head. We first study the effect of the scaling transformation in box-attention on feature pyramids as it could possibly generate regions-of-interest at different scales. More specifically, we compare the area between generated regions and the initial reference window of query vectors corresponding to object proposals in the last encoder layer. Surprisingly, the change of area after scaling has a mean of 31% and standard deviation 33% w.r.t. the original area of the reference window (*e.g.*, for a reference window of $32 \times 32$ pixels to capture regions of different scales, such as $64 \times 64$ or $16 \times 16$ pixels, the change of the area should be more than 75%). This suggests that the scaling function in box-attention prefers to capture regions of different aspect ratios rather than multi-scale information. Fig. 1a further confirms that feature pyramids are important for current detectors to detect objects of various sizes. It can be seen in Fig. 1b that deploying feature pyramids brings a large improvement to different types of detectors (*i.e.*, ~2 AP points for BoxeR and ~3 AP points for ViTDet). This observation is consistent to the observation in DeformableDETR (Zhu et al., 2021) with multi-scale deformable attention on hierarchical backbones.

**SimPLR is an effective single-scale detector.** In Tab. 1, we show the comparison between SimPLR and recent object detectors using the plain backbone ViT. The plain detector, SimPLR, removes the need for multi-scale adaptation of the ViT. We report ViTDet with the setting in Li et al. (2022); and BoxeR along with SimPLR using the $5\times$ schedule as in Nguyen et al. (2022). Tab. 1 indicates that SimPLR with single-scale input outperforms ViTDet and reaches competitive performance compared to BoxeR. This is different from recent object detectors with transformers (Zhu et al., 2021; Cheng et al., 2022) in which multi-scale feature maps are needed to achieve strong performance for both object detection and segmentation. Also, even with the complicated adaption of the ViT backbone, UViT (Chen et al., 2022) with single-scale features still shows worse performance compared to ViTDet that deploys feature pyramids. In contrast, our experiment demonstrates that single-scale features are sufficient and the proposed design of SimPLR allows features to learn suitable scale information for detecting objects of various scales. We find that the significance of multi-scale feature maps diminishes when the encoder is equipped with a powerful attention mechanism.

**Ablation of scale-aware attention.** For this ablation on the importance of scale-aware attention in the design of SimPLR, our baseline is the original box-attention from Nguyen et al. (2022) with single-scale feature input directly taken from the last feature of the ViT backbone (denoted as "base").

From Tab. 2a we first conclude that *both* scale-aware attention strategies are substantially better than the naïve baseline, increasing AP by up to 1.8 points. We note that while fixed-scale attention distributes 25% of its attention heads into each of the window scales, adaptive-scale attention decides the scale distribution based on the query content. By choosing feature grids from different window

| | FLOPs | FPS | AP$^b$ | AP$^m$ |
|---|---|---|---|---|
| **Feature pyramids** | | | | |
| ViTDet | 1.1T | 11 | 54.0 | 46.7 |
| BoxeR | 0.6T | 12 | 55.4 | 47.7 |
| **Single-scale** | | | | |
| SimPLR | 0.5T | 15 | 55.4 | 47.6 |

Table 1: **SimPLR is an effective single-scale detector.** All methods use ViT-B as backbone. Both ViTDet and BoxeR employ SimpleFPN with ViT from Li et al. (2022). Our single-scale detector, SimPLR, shows competitive performance compared to multi-scale alternatives, while being more efficient in terms of FLOPs and faster during inference. Notably, SimPLR can reach 25 frames-per-second with jit optimization.

| attention | AP$^b$ | AP$^m$ |
|---|---|---|
| base | 53.6 | 46.1 |
| i) fixed-scale | 55.0 | 47.2 |
| ii) adaptive-scale | 55.4 | 47.6 |
|  |  |  |

(a) **Scale-aware attention.**

| $s$ | AP$^b$ | AP$^m$ |
|---|---|---|
| base | 53.6 | 46.1 |
| 16 | 55.1 | 47.4 |
| 32 | 55.4 | 47.6 |
| 64 | 55.1 | 47.4 |

(b) **Window size.**

| $n$ | AP$^b$ | AP$^m$ |
|---|---|---|
| base | 53.6 | 46.1 |
| 2 | 54.6 | 47.0 |
| 4 | 55.4 | 47.6 |
| 6 | 55.2 | 47.6 |

(c) **Number of window scales.**

| scale | AP$^b$ | AP$^m$ |
|---|---|---|
| base | 53.6 | 46.1 |
| 1/4 | 55.4 | 47.7 |
| 1/8 | 55.4 | 47.6 |
| 1/16 | 54.3 | 46.7 |

(d) **Scales of input features.**

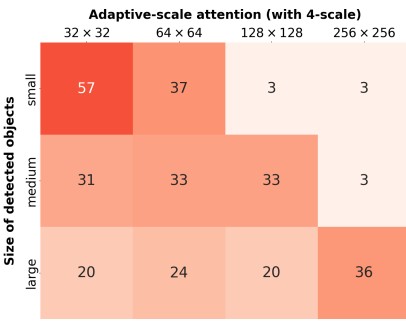

(e) Visualization of scale distribution learnt in **multi-head adaptive-scale attention** of object proposals. Objects are classified into *small*, *medium*, and *large* based on their area.

Table 2: **Ablation of scale-aware attention** in SimPLR using a plain ViT backbone on COCO `val`. **Table (a-d):** Compared to the naïve baseline, which employs BoxeR and box-attention (Nguyen et al., 2022) with *single-scale* features, our plain detector, SimPLR, with scale-aware attention improves the performance consistently for all settings, best one highlighted. **Figure e:** our adaptive-scale attention captures different scale distribution in its attention heads based on the context of query vectors. Specifically, queries of *small* objects tends to focus on reference windows of small scales (*i.e.*, mainly $32 \times 32$), while query vectors of *medium* and *large* objects distribute more attention computation into larger reference windows.

scales adaptively, the adaptive-scale attention is able to learn a suitable scale distribution through training data, yielding better performance compared to fixed-scale attention. This is also verified in Tab. 2e where queries corresponding to *small* objects tend to pick reference windows of small sizes for its attention heads. Interestingly, queries corresponding to *medium* and *large* objects pick not only reference windows of their sizes, but also ones of smaller sizes. One of reasons may come from the fact that performing instance segmentation of larger objects still requires the network to faithfully preserve the per-pixel spatial details.

In Tab. 2b we compare the performance of SimPLR across several sizes ($s$) of the reference window. They all improve over the baseline, while the choice of a specific base size makes only marginal differences. Our ablation reveals that the number of scales rather than the window size plays an important role to make our network more *scale-aware*. Indeed, in Tab. 2c, the use of 4 or more window scales shows improvement up to 0.8 AP over 2 window scales; and clearly outperforms the naïve baseline. Last, we show in Tab. 2d that the *decouple* between feature scale and dimension of the ViT backbone and the detection head features helps to boost the performance of our plain detector by ∼1 AP point, while keeping its efficiency (*i.e.*, both in terms of FLOPs and FPS). This practice makes scaling of SimPLR to larger ViT backbones more practical.

**State-of-the-art comparison and scaling behavior.** We show in Tabs. 3 and 4 that SimPLR indicates strong performance on object detection, instance segmentation and panoptic segmentation. Specifically, our plain detector combined with a ViT, pre-trained using MAE (He et al., 2022) or BEiTv2 (Peng et al., 2022), presents good scaling behavior. When moving to large and huge models, our method outperforms multi-scale counterparts including the recent end-to-end Mask2Former

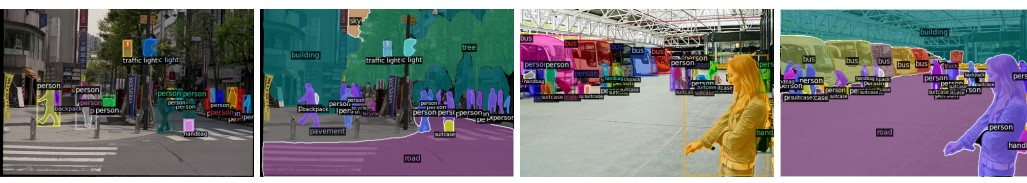

Figure 3: **Qualitative results** for object detection, segmentation and panoptic segmentation on COCO `val` as generated by SimPLR with ViT-B backbone. SimPLR can detect and segment objects in crowded scenes.

| method | backbone | pre-train | Object Detection | | | | Instance Segmentation | | | | FPS |
|---|---|---|---|---|---|---|---|---|---|---|---|
| | | | $AP^b$ | $AP^b_S$ | $AP^b_M$ | $AP^b_L$ | $AP^m$ | $AP^m_S$ | $AP^m_M$ | $AP^m_L$ | |
| **Base models** | | | | | | | | | | | |
| Swin | Swin-B | sup-22K | 54.0 | - | - | - | 46.5 | - | - | - | 13 |
| Mask2Former | Swin-B | sup-22K | n/a | | | | **48.1** | **27.8** | 52.0 | **71.1** | - |
| MViT | MViT-B | sup-22K | 55.6 | - | - | - | **48.1** | - | - | - | 11 |
| BoxeR | ViT-B | MAE | 55.4 | - | - | - | 47.7 | - | - | - | 12 |
| ViTDet | ViT-B | MAE | 54.0 | 36.5 | 57.8 | 69.1 | 46.7 | 27.2 | 49.6 | 64.9 | 11 |
| UViT | UViT-B | self-learning | 53.9 | - | - | - | 46.1 | - | - | - | 12 |
| SimPLR | ViT-B | MAE | 55.6 | **37.1** | 59.2 | 71.5 | 48.0 | 27.5 | 51.5 | 67.8 | **15** |
| SimPLR | ViT-B | BEiTv2 | **55.7** | 36.5 | **60.2** | **72.3** | **48.1** | 26.7 | **52.7** | 68.9 | **15** |
| **Large models** | | | | | | | | | | | |
| Swin | Swin-L | sup-22K | 54.8 | - | - | - | 47.3 | - | - | - | **10** |
| Mask2Former | Swin-L | sup-22K | n/a | | | | 50.1 | 29.9 | 53.9 | **72.1** | 4 |
| MViT | MViT-L | sup-22K | 55.7 | - | - | - | 48.3 | - | - | - | 6 |
| ViTDet | ViT-L | MAE | 57.6 | 40.5 | 61.6 | 72.6 | 49.9 | 30.5 | 53.3 | 68.0 | 7 |
| SimPLR | ViT-L | MAE | 58.3 | **42.2** | 62.3 | 73.1 | 50.4 | **32.0** | 54.3 | 69.5 | 9 |
| SimPLR | ViT-L | BEiTv2 | **58.5** | 40.1 | **63.4** | **74.6** | **50.7** | 30.2 | **55.3** | 70.8 | 9 |
| **Huge models** | | | | | | | | | | | |
| MViT | MViT-H | sup-22K | 55.7 | - | - | - | 48.3 | - | - | - | 6 |
| ViTDet | ViT-H | MAE | 58.7 | **41.9** | 63.0 | 73.9 | 50.9 | **32.0** | 54.3 | 68.9 | 5 |
| SimPLR | ViT-H | MAE | **59.5** | 41.8 | **63.5** | **75.0** | **51.6** | 31.7 | **55.6** | 70.9 | **7** |

Table 3: **State-of-the-art comparison and scaling behavior for object detection and instance segmentation.** We compare between methods using feature pyramids (*top* row) *vs.* single-scale (*bottom* row) on COCO (Lin et al., 2014) `val`. Backbones with MAE pre-trained on ImageNet-1K while others pre-trained on ImageNet-22K. Methods in gray color are with convolution-based detection head. (n/a: entry is not available; BEiTv2 uses intermediate finetuning with ImageNet-22K). Models of larger sizes are with *darker* orange color. SimPLR indicates good scaling behavior. With only single-scale features, SimPLR shows strong performance compared to multi-scale detectors including transformer-based detectors like Mask2Former, while being faster.

| method | backbone | pre-train | Panoptic Segmentation | | | FPS |
|---|---|---|---|---|---|---|
| | | | PQ | $PQ^{th}$ | $PQ^{st}$ | |
| **Base models** | | | | | | |
| MaskFormer | Swin-B | sup-22K | 51.8 | 56.9 | 44.1 | - |
| Mask2Former | Swin-B | sup-22K | 56.4 | 62.4 | **47.3** | - |
| SimPLR | ViT-B | BEiTv2 | **56.5** | **62.6** | **47.3** | 13 |
| **Large models** | | | | | | |
| Mask2Former | Swin-L | sup-22K | 57.8 | 64.2 | 48.1 | 4 |
| SimPLR | ViT-L | BEiTv2 | **58.5** | **65.1** | **48.6** | **8** |

Table 4: **State-of-the-art comparison and scaling behavior for panoptic segmentation.** We compare between methods using feature pyramids (*top* row) *vs.* single-scale (*bottom* row) on COCO `val`. All backbones are pre-trained on ImageNet-22K. Models of larger sizes are with *darker* orange color. SimPLR shows better results when scaling to larger backbones, while being faster with single-scale input.

segmentation model (Cheng et al., 2022). Despite involving more advanced attention blocks designs, *i.e.*, shifted window attention in Swin (Liu et al., 2021) and pooling attention in MViT (Fan et al., 2021), detectors with hierarchical backbones benefit less from larger backbones. SimPLR is better than ViTDet across all backbones in terms of both accuracy and inference speed. The visualization of the SimPLR prediction can be seen in Fig. 3 (more visualizations are provided in the supplementary).

**Limitations.** Our final goal is to simplify the detection pipeline and to achieve competitive results at the same time. In Sec. 4, we find that the *adaptive-scale* attention mechanism that adaptively learns scale-aware information in its computation plays a key role for a plain detector. However, our adaptive-scale attention still encodes the knowledge of different scales. In the future, we hope that with the help of large-scale training data, an even simpler design of the attention mechanism could also learn the scale equivariant property. Furthermore, SimPLR faces difficulties in detecting and segmenting large objects in the image. To overcome this limitation, we think that a design of attention computation which effectively combines both global and local information is necessary.

## 5 RELATED WORK

**Backbones for object detection.** Inspired by R-CNN (Girshick et al., 2014), modern object detection methods utilize a task-specific head on top of a pre-trained backbone. Initially, object detectors were dominated by convolutional neural network (CNN) backbones (LeCun & Bengio, 1995) pre-trained on ImageNet (Deng et al., 2009), with consistent accuracy and efficiency improvements over subsequent generations of architectures, *e.g.* (Simonyan & Zisserman, 2015; Xie et al., 2017; He et al., 2016; Huang et al., 2017). With the success of the transformer in learning from large-scale text data (Brown et al., 2020; Devlin et al., 2019), many studies have explored the transformer for computer vision (Chen et al., 2020a; Dosovitskiy et al., 2021). By removing the need for labels, methods with self-supervised learning have emerged as an even more powerful solution for pre-training general vision representations (Chen et al., 2020b; He et al., 2022). The Vision Transformer (ViT) (Dosovitskiy et al., 2021) demonstrated the capability of learning meaningful representation for visual recognition. We show through experiments that the pre-trained ViT when combined with our proposed encoder-decoder delivers strong performance in detection and segmentation while removing non-trivial adaptations in ViT for feature pyramids.

**Multi-scale object detection.** A key challenge in object detection is to detect objects within an image across multiple scales. Early works tackled this problem by applying a CNN detector with a sliding window strategy on an image pyramid (Sermanet et al., 2014) or with generated region proposals (Uijlings et al., 2013) to extract each scale-normalized region from the input image (Girshick et al., 2014). To reduce computational costs, Faster R-CNN (Ren et al., 2015) generates object proposals on the feature map using a region proposal network. As deeper features of CNNs tends to capture more high-level information at the expense of fine-grained details for small objects, SSD (Liu et al., 2016) predicts objects from multiple layers of the feature hierarchy. The Feature Pyramid Network (Lin et al., 2017) further improves the creation of multi-scale feature maps with top-down fusion and lateral connections. With the recent evidence that non-hierarchical transformer architectures (*i.e.*, ViTs) are able to learn convolution-like behaviour through image data (*i.e.*, translation-equivariance), the necessity of multi-scale feature maps becomes questionable. In this study, we demonstrate that a plain ViT backbone along with a single-scale encoder-decoder is able to detect multi-scale objects without the need for feature pyramids.

**End-to-end detection and segmentation.** A universal architecture solves multiple vision tasks without many architectural changes. This is made possible by the introduction of transformers for object detection (Carion et al., 2020). Follow-up works (Dong et al., 2021; Wang et al., 2021a; Nguyen et al., 2022) extended the end-to-end detection framework to object detection, instance segmentation, and panoptic segmentation. This inspired MaskFormer (Cheng et al., 2021) and K-Net (Zhang et al., 2021) to unify segmentation tasks with a class-agnostic mask prediction. Pointing out that MaskFormer and K-Net still lag behind specialized architectures, Cheng et al. (2022) introduce Mask2Former, which outperforms specialized architectures on instance, semantic, and panoptic segmentation. Yu et al. (2022) replace self-attention with $k$-means clustering, further boosting the effectiveness of the network. Another direction is to improve the object query in the decoder via a denoising process (Zhang et al., 2023; Li et al., 2023). Similar to convolution-based detection heads, these approaches utilize multi-scale feature maps from a hierarchical backbone. In this work, we remove the multi-scale feature map constraint and enable SimPLR on detection and segmentation using a *single-scale* feature map.

## 6 CONCLUSION

We presented SimPLR, a simple and plain object detector that eliminates the requirement for handcrafting multi-scale feature maps. Through our experiments, we demonstrated that a transformer-based detector, equipped with a scale-aware attention mechanism, can effectively learn scale-equivariant features through data. The efficient design of SimPLR allows it to take advantages of significant progress in scaling ViTs, reaching highly competitive performance on three tasks on COCO: object detection, instance segmentation, and panoptic segmentation. This finding suggests that many hand-crafted designs for convolution neural network in computer vision could be removed when moving to transformer-based architecture. We hope this study could encourage future exploration in simplifying neural network architectures especially for dense vision tasks.

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
