# SimPLR: A Simple and Plain Transformer for Object Detection and Segmentation —Supplementary Material—

## Abstract

This supplementary material provides additional implementation details, further information for better reproducibility, additional quantitative and qualitative results as well as license information.

## Contents

## A   Implementation Details

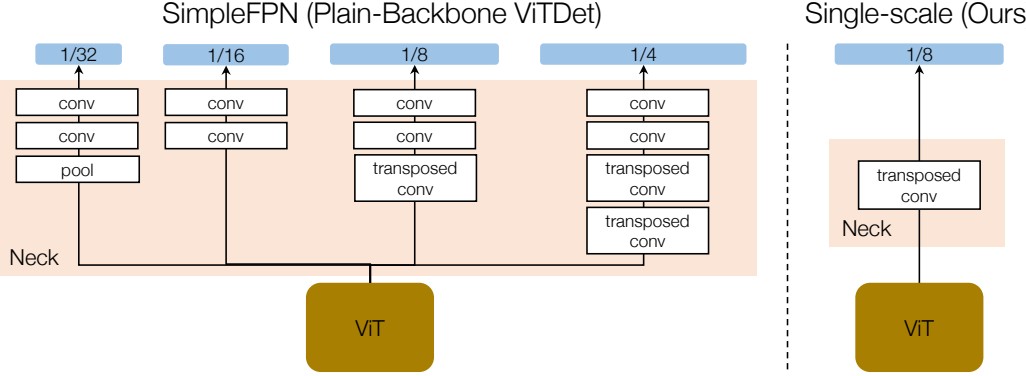

Figure A: **The creation of input features. Left:** The creation of feature pyramids from the last feature of the plain backbone, ViT, in SimpleFPN (Li et al., 2022) where different stacks of convolutional layers are used to create features at different scales. **Right:** The design of our single-scale feature map with only one layer.

**The creation of input features.**  In Fig. A, we compare the creation of input features to detection head between SimpleFPN and our method. In Li et al. (2022), the multi-scale feature maps are created by different sets of convolution layers. Instead, SimPLR simply applies a deconvolution layer following by a GroupNorm layer (Wu & He, 2018).

**Masked Instance-Attention.**  The masked instance-attention follows the grid sampling strategy of the box-attention in Nguyen et al. (2022), but differs in the computation of attention scores to better capture objects of different shapes. To be specific, the region of interest $r'_i$ is divided into 4 bins of $2 \times 2$ grid, each of which contains a $\frac{m}{2} \times \frac{m}{2}$ grid features sampled using bilinear interpolation. Instead of assigning an attention weight to each feature vector, a linear projection ($\mathbb{R}^d \to \mathbb{R}^{2 \times 2}$) is adopted to generate the $2 \times 2$ attention scores for 4 bins. The $\frac{m}{2} \times \frac{m}{2}$ feature vectors within the same

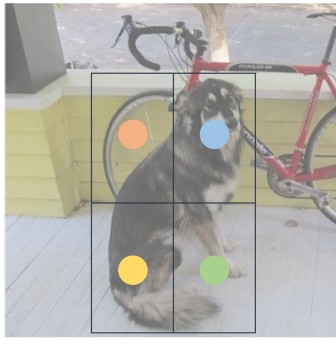 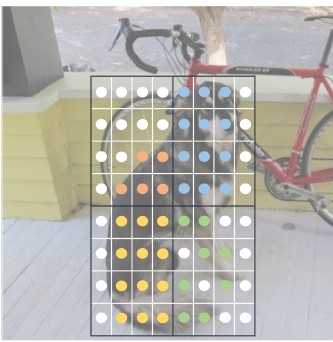

Figure B: **Masked Instance-Attention. Left:** The box-attention (Nguyen et al., 2022) which samples $2 \times 2$ grid features in the region of interest. **Right:** Our masked instance-attention for dense grid sampling that employs masking strategy to capture object boundary. The $2 \times 2$ attention scores are denoted in four colours and the masked attention score is shown in white.

bin share the same attention weight. This is equivalent to the *average* aggregation of feature values covered by each bin, which shows to reduce misalignments in RoIAlign (He et al., 2017):

$$\text{head}_i = \sum_{k=0}^{2 \times 2} \sum_{j=0}^{\frac{m}{2} \times \frac{m}{2}} \frac{\alpha_k}{\frac{m}{2} \cdot \frac{m}{2}} \, v_{i_{k,j}}, \tag{1}$$

where $a_k$ is the attention weight corresponding to $k$-th bin and $v_{i_{k,j}}$ is the $j$-th feature vector inside $k$-th bin.

Inspired by Cheng et al. (2022), we utilize the mask prediction of the previous decoder layer $\mathcal{M}_q \in \mathbb{R}^{H_m \times W_m}$ corresponding to the object query $q$. Given the coordinates of grid features within the region of interest $r'_i$, we sample the corresponding mask scores using bilinear interpolation. The sampled mask scores are binarized with the 0.5 threshold before $\text{softmax}$ in the attention computation. Note that in masked instance-attention, we sample the feature grid of $14 \times 14$.

Fig. B shows the difference between box-attention (Nguyen et al., 2022) and masked instance-attention. By utilizing the mask prediction from previous decoder layer, masked instance-attention can effectively capture object of different shapes.

**Losses in training of SimPLR.** We use focal loss (Lin et al., 2017) and dice loss (Milletari et al., 2016) for the mask loss: $\mathcal{L}_{\text{mask}} = \lambda_{\text{focal}} \mathcal{L}_{\text{focal}} + \lambda_{\text{dice}} \mathcal{L}_{\text{dice}}$ with $\lambda_{\text{focal}} = \lambda_{\text{dice}} = 5.0$. The box loss is the combination of $\ell_1$ loss and GIoU loss (Rezatofighi et al., 2019), $\mathcal{L}_{\text{box}} = \lambda_{\ell_1} \mathcal{L}_{\ell_1} + \lambda_{\text{giou}} \mathcal{L}_{\text{giou}}$, with $\lambda_{\ell_1} = 5.0$ and $\lambda_{\text{giou}} = 2.0$. The focal loss is also used for our classification loss, $\mathcal{L}_{\text{cls}}$. Our final loss is formulated as: $\mathcal{L} = \mathcal{L}_{\text{mask}} + \mathcal{L}_{\text{box}} + \lambda_{\text{cls}} \mathcal{L}_{\text{cls}}$ ($\lambda_{\text{cls}} = 2.0$ for object detection and instance segmentation, $\lambda_{\text{cls}} = 4.0$ for panoptic segmentation).

**Hyper-parameters of SimPLR.** SimPLR contains 6 encoder and decoder layers. The adaptive-scale attention in SimPLR encoder samples $2 \times 2$ grid features per region of interest. In the decoder, we compute attention on a grid of $14 \times 14$ features within regions of interest. The dimension ratio of feed-forward sub-layers to 4. The number of object queries is 300 in the decoder as suggested in Nguyen et al. (2022). The size of input image is $1024 \times 1024$ in both training and inference. Note that we also use this setting for the baseline (*i.e.*, BoxeR with ViT backbone).

In Tab. 2d, we show that the *decouple* between feature scale and dimension of the ViT backbone and the detection head helps to boost the performance of our plain detector while keeping the efficiency. This comes from the fact that the complexity of global self-attention in the ViT backbone increase quadraticaly w.r.t. the feature scale and the detection head enjoys the high-resolution input for object prediction. Note that with ViT-H as the backbone, we follow Li et al. (2022) to interpolate the kernel of patch projection into $16 \times 16$. The hyper-parameters for each SimPLR size (Base, Large, and Huge) are in Tab. A.

## B ADDITIONAL RESULTS

**More panoptic segmentation comparison.** Here, we provide more results of SimPLR with ViT-B pre-trained using MAE on COCO panoptic segmentation in Tab. B. SimPLR with MAE pre-trained backbone continues to show strong segmentation performance when using only single-scale input.

| model size | backbone | | | detection head | | | |
|---|---|---|---|---|---|---|---|
| | dim. | # heads | feature scale | encoder dim. | decoder dim. | # heads | feature scale |
| Base | 768 | 12 | $\frac{1}{16}$ | 384 | 256 | 12 | $\frac{1}{8}$ |
| Large | 1024 | 16 | $\frac{1}{16}$ | 768 | 384 | 16 | $\frac{1}{8}$ |
| Huge | 1280 | 16 | $\frac{1}{16}$ | 960 | 384 | 16 | $\frac{1}{8}$ |

Table A: Hyper-parameters of backbone and detection head for different sizes of SimPLR (base – large – huge models). Note that these settings are the same for all three tasks.

| method | backbone | pre-train | Panoptic Segmentation | | | FPS |
|---|---|---|---|---|---|---|
| | | | PQ | $PQ^{th}$ | $PQ^{st}$ | |
| MaskFormer | Swin-B | sup-1K | 51.1 | 56.3 | 43.2 | - |
| Mask2Former | Swin-B | sup-1K | 55.1 | 61.0 | **46.1** | - |
| SimPLR | ViT-B | MAE | **55.3** | **61.6** | 45.8 | **13** |

Table B: **More panoptic segmentation comparison** between SimPLR with ViT-B backbone pre-trained using MAE and other methods with Swin-B backbone. All backbones are pre-trained on ImageNet-1K. SimPLR still shows competitive results when using only single-scale input.

**Ablation on pre-training strategies.** Tab. C compares the ViT backbone when pre-trained using different strategies with different sizes of pre-training data. SimPLR with the ViT backbone benefits from better pre-training methods even with supervised approaches. Specifically, ViT pre-trained with DEiTv3 (Touvron et al., 2022) is better than one with DEiT (Touvron et al., 2021), and the pre-training on ImageNet-21K further improves the performance of DEiTv3.

However, the self-supervised method like MAE (He et al., 2022) provides strong pre-trained backbones when only pre-trained on ImageNet-1K. This further confirms that our plain detector, SimPLR, enjoys the significant progress of self-supervised learning and scaling ViTs. A similar observation is also pointed out in ViTDet (Li et al., 2022) where the ViT backbone initialized with MAE improves the upper-bound of object detection compared to the long and optimal training recipe from scratch.

| pre-train | Object Detection | | | | Instance Segmentation | | | |
|---|---|---|---|---|---|---|---|---|
| | $AP^b$ | $AP_S^b$ | $AP_M^b$ | $AP_L^b$ | $AP^m$ | $AP_S^m$ | $AP_M^m$ | $AP_L^m$ |
| IN-1K, DEiT | 53.6 | 33.7 | 58.1 | 71.5 | 46.1 | 24.5 | 50.4 | 67.2 |
| IN-1K, DEiTv3 | 54.0 | 34.3 | 58.8 | 70.5 | 46.4 | 24.8 | 51.1 | 66.7 |
| IN-21K, DEiTv3 | 54.8 | 35.4 | 59.0 | **72.4** | 47.1 | 25.8 | 51.2 | **68.5** |
| IN-1K, MAE | **55.4** | **36.1** | **59.1** | 70.9 | **47.6** | **26.8** | **51.4** | 67.1 |

Table C: **Ablation on pre-training strategies** of the plain ViT backbone using SimPLR evaluated on COCO object detection and instance segmentation. We compare the ViT backbone pre-trained using supervised methods (*top* row) *vs.* self-supervised methods (*bottom* row) with different sizes of pre-training dataset (ImageNet-1K *vs.* ImageNet-21K). Here, we use the $5\times$ schedule as in Nguyen et al. (2022). It can be seen that SimPLR with the plain ViT backbone benefits from better pre-training approaches and with more pre-training data.

## C   QUALITATIVE RESULTS

We provide qualitative results of the SimPLR prediction with ViT-B backbone on three tasks: COCO object detection, instance segmentation, and panoptic segmentation in Fig. C.

## D   ASSET LICENSES

| Dataset | License |
|---|---|
| ImageNet (Deng et al., 2009) | https://image-net.org/download.php |
| COCO (Lin et al., 2014) | Creative Commons Attribution 4.0 License |

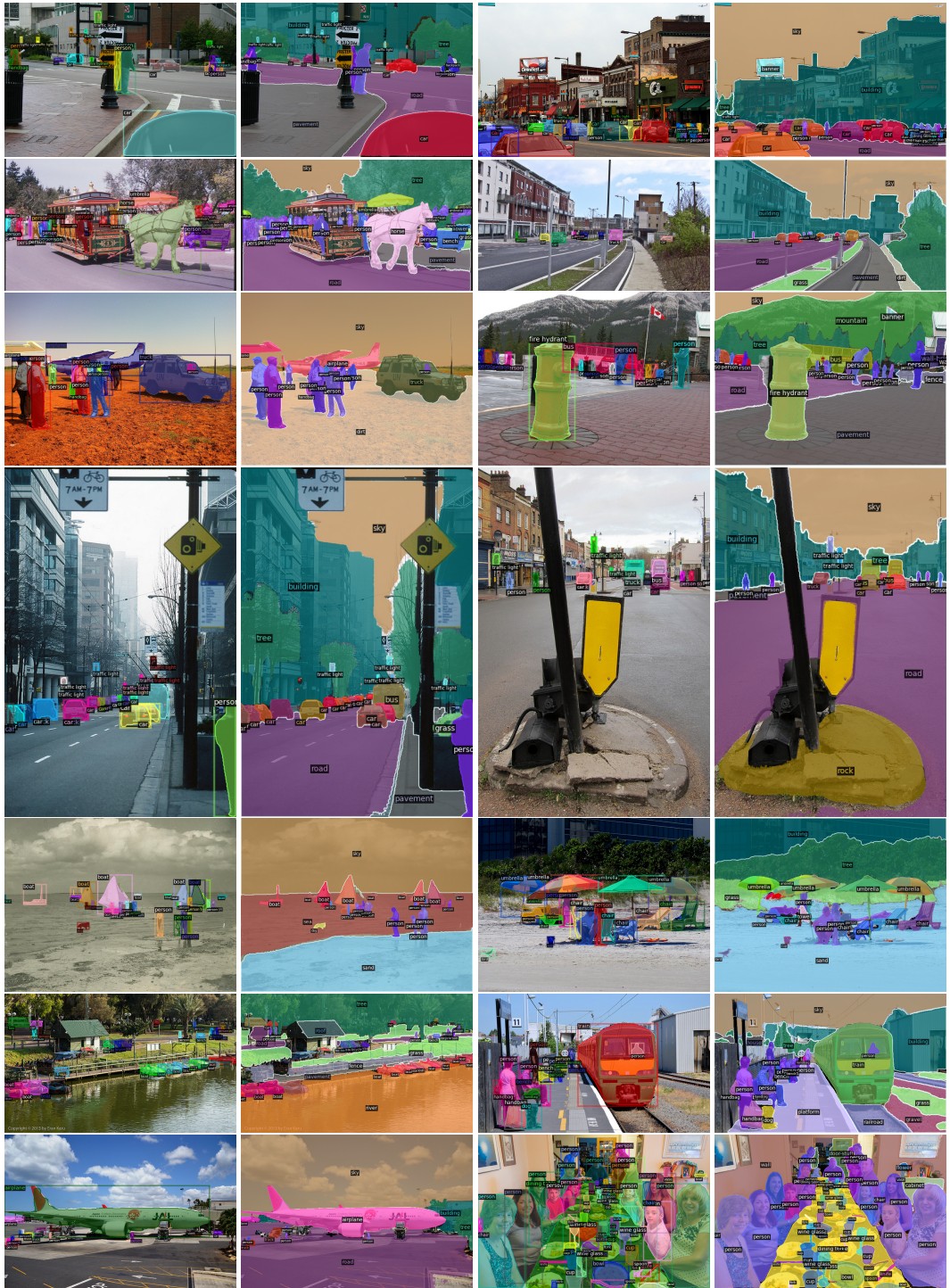

Figure C: **Qualitative results** for object detection, instance segmentation, and panoptic segmentation generated by SimPLR using ViT-B as backbone on the COCO `val` set. In each pair, the left image shows the visualization of object detection and instance segmentation, while the right one indicates the panoptic segmentation prediction.