# OpenReview forum: "SimPLR: A Simple and Plain Transformer for Object Detection and Segmentation"
_ICLR.cc/2024/Conference — ICLR 2024 Conference Withdrawn Submission_

### Official Review · Reviewer_SZaB · 2023-10-29

**Soundness:** 3 good
**Presentation:** 3 good
**Contribution:** 2 fair
**Rating:** 3
**Confidence:** 4

**Summary:**

This paper presents SimPLR. It shows that a plain vision transformer without feature pyramids but with scale-aware attention is able to achieve detection and segmentation performance comparable with pyramid designs.

**Strengths:**

+ The presented SimPLR detector maintain the design philosophy of simplicity in ViTs, with a plain architecture and single-scale features.
+ SimPLR indicates a clear advantage over the ViTDet baseline.
+ SimPLR reports good performance on three dense prediction tasks on COCO.

**Weaknesses:**

- The novelty of the work seems marginal. A very similar observation has been made recently. See (DETR Does Not Need Multi-Scale or Locality Design, ICCV’23). Learning an object detector with single-scale features is also not new. See (You Only Look One-Level Feature, CVPR’21)

- The proposed multi-head scale-aware attention is an incremental extension of the Box Attention by (Nguyen et al., 2022). The extension from the Box Attention to the Fixed-Scale Attention and Adaptive-Scale Attention is straightforward. This leads to rather limited technical contribution.

- The improvement over the BoxeR is marginal. In Table 1, SimPLR achieves the same performance as BoxeR (55.4 box AP), with also comparable FLOPs. While SimPLR is slightly efficient, the improvement is not significant.

- The claim of SimPLR outperforms the multi-scale Mask2Former is questionable. In Table 4, the comparison is unfair. SimPLR and Mask2Former use different backbones and pretraining. Such a comparison seems meaningless.

**Questions:**

I don't have additional questions for the authors. The paper is easy to understand, and the problem addressed is clear.

While I appreciate the simplicity of the SimPLR and the substantial experiments performed, the results and the claim are not surprising enough or can find similar work in open literature. The technical contribution is also somewhat incremental.

---

> ### Author Response · Authors · 2023-11-17
>
> Thanks for the detailed and constructive comments to help our work, acknowledging the ```simplicity``` of our approach.
>
> Our response to each concern:
> 1. ***The novelty of the work seems marginal. A very similar observation has been made recently. See (DETR Does Not Need Multi-Scale or Locality Design, ICCV’23). Learning an object detector with single-scale features is also not new. See (You Only Look One-Level Feature, CVPR’21)***
>
> - Our goal is to pursue the plain detector for both detection and segmentation. In YOLOF [a], while the authors use single-scale input in the detection head, the backbone is still hierarchical. In PlainDETR [b], the multi-scale features are still necessary to generate object proposals. Instead, our SimPLR completely removes the hierarchical and multi-scale constraints in both backbone and encoder of the network.
> - Furthermore, it can be seen that our work is a natural step towards the plain detector. While YOLOF simplifies the detection pipeline by using single-scale input for the detection head, the PlainDETR takes a step forward by removing the multi-scale input in backbone and encoder (even though, they still need multi-scale features for proposal generation). In our work, we push the direction of plain detector forward by completely removing these constraints. Moreover, our plain detector can perform both detection and segmentation
>
> 2. ***The proposed multi-head scale-aware attention is an incremental extension of the Box Attention by (Nguyen et al., 2022)***
>
> - We choose to find a simple but effective approach rather than a more complicated one. This is proved by the performance of our detector (SimPLR outperforms PlainDETR by 1.6 AP point and can perform segmentation as well). As mentioned above, we believe that it is much harder to find a simple and effective solution.
> - Our work gives an insight about learning scale-equivariant features in attention computation and a simple setting of plain detector. We strongly believe that the contribution of our work is valuable to the community.
>
> 3. ***The improvement over the BoxeR is marginal.**
>
> - Similar to YOLOF or PlainDETR, the goal of our work is to simplify the detection pipeline. Regarding the performance, our plain detector shows better performance than PlainDETR. It also shows a good scaling behaviour while being more efficient. This is different from multi-scale detector like Mask2Former when the speed is very slow with larger backbones.
>
> 4. ***The claim of SimPLR outperforms the multi-scale Mask2Former is questionable. In Table 4, the comparison is unfair. SimPLR and Mask2Former use different backbones and pretraining. Such a comparison seems meaningless.***
>
> - It is clear that our detector shows 2 times faster speed than Mask2Former. Furthermore, we also acknowledge that SimPLR benefits from the self-supervised learning and scaling research progress with ViT. We think it's the advantages of using plain backbone like ViT. Similarly, in ViTDet, the authors show that ViT seems to benefit more from self-supervised learning than hierarchical backbones.
>
> [a] Chen et al. You Only Look One-level Feature. In CVPR 2021.
>
> [b] Lin et al. DETR Doesn't Need Multi-Scale or Locality Design. In ICCV 2023.

---

### Official Review · Reviewer_ogBT · 2023-10-31

**Soundness:** 3 good
**Presentation:** 3 good
**Contribution:** 2 fair
**Rating:** 5
**Confidence:** 3

**Summary:**

This paper presents SIMPLR, a simple and straightforward transformer for object detection and segmentation. SIMPLR aims to eliminate the need for the Feature Pyramid Network structure and instead introduces scale-aware attention. This allows the backbone and detection head to effectively utilize single-scale features. Extensive experiments show that SIMPLR outperforms other detectors equipped with FPN while maintaining superior speed.

**Strengths:**

1. The motivation behind the proposed components, scale-aware attention, is clear and effectively addresses the issue of relying on FPN.

2. By introducing the idea of cropping box features at different scales from a single-scale map, similar performance to FPN can be achieved. The idea is logical and makes sense in this context.

3. The experiments conducted in this study are robust and thorough. The author systematically evaluates the impact of various hyper-parameters and settings of the proposed multi-scale box attention method. Furthermore, the author compares the performance of the proposed method with state-of-the-art approaches on different datasets and tasks.

**Weaknesses:**

1. If I've understood correctly, the primary contribution of this article is the concept of scale-aware attention. However, I noted the absence of any graphical representation to thoroughly elucidate this complex concept. It would be immensely beneficial if a detailed, comprehensible image could be included to help readers better grasp the intricate technicalities of this key component.

2. While the paper presents some novel aspects, there are areas that fall short. From my understanding, the key innovation lies in the scale-aware attention, which builds upon box-attention [1] by assembling attention from boxes at different scales. While I appreciate this advancement, I believe it may not be sufficient on its own. I fear that it may not meet the rigorous standards expected of an ICLR paper.

3. While the author presents the performance of other methods utilizing single-scale feature maps in Figure 1, it might be more compelling to incorporate detailed, quantified comparisons in the Experiments Section. This would further validate the effectiveness of SIMPLR.

[1] BoxeR: Box-Attention for 2D and 3D Transformers

**Questions:**

I have stated my questions and doubts in the weakness section. Please correct me if I am wrong.

---

> ### Author Response · Authors · 2023-11-17
>
> Thanks for the detailed and constructive comments to help our work, acknowledging our approach: ```maintaining superior speed```, ```idea is logical```, ```experiments conducted are robust and thorough```:
>
> Our response to each concern:
> 1. ***If I've understood correctly, the primary contribution of this article is the concept of scale-aware attention.***
>
> - The primary contribution of our work is to show that a plain detector can achieve strong performance compared to multi-scale counterparts in both object detection and segmentation. This is challenging problem specially for segmentation.
>
> - We show that the plain detector is made possible by: 1) the scale-aware attention that incorporates scale information into attention computation for scale-equivariant learning (this is proven in our ablation), 2) our proposed detector is highly efficient and shows a good scaling behaviour. We believe our work is significant as our method is scalable and shows good performance
>
> 2. ***While the paper presents some novel aspects, there are areas that fall short. From my understanding, the key innovation lies in the scale-aware attention, which builds upon box-attention [1] by assembling attention from boxes at different scales. While I appreciate this advancement, I believe it may not be sufficient on its own.***
>
> - While our method is simple, it is efficient and effective. We think that it's more difficult to find a simple and effective method and choose to not go for more complicated settings.
> - This applies to the NLP as well when global self-attention is expensive when dealing with long sequence input, they need to come up with sparse attention [a] to tackle the problem. This is even more problematic in detection and segmentation due to high-resolution images.
>
> [a] Qiu et al. Blockwise Self-Attention for Long Document Understanding. In EMNLP 2020.

---

### Official Review · Reviewer_VYm9 · 2023-11-01

**Soundness:** 3 good
**Presentation:** 3 good
**Contribution:** 3 good
**Rating:** 6
**Confidence:** 5

**Summary:**

The paper proposes an object detector that is based on single-scale plain ViT backbone. It follows the detector of DETR and adapts scale-aware attention for learning multi-scale features from single-scale backbone features for detection. Experiments are conducted on COCO to show the performance of both detection and segmentation. SimPLR outperform ViTDet and Mask2Former with single-scale backbone (plain ViT) on detection and segmentation task.

**Strengths:**

The motivation and exploration of using single-scale feature / backbone for object detection is highly meaningful. Previous attempt like ViTDet tries to keep the pretrained backbone plain, but still needs to produce feature pyramids for classic object detectors to perform well.  SimPLR uses multi-scale attention instead to get comparable performance. It extends box-attention with multiple reference windows of different scales, similar to RPN.

**Weaknesses:**

While SimPLR is able to get comparable/better performance with ViTDet, it is still behind sota object detector (e.g. DINO), it is not clear how SimPLR can be transferred to different advanced detectors.

**Questions:**

The proposed method is based on a specific box-attention approach,  can such method be applied to other transformer-based detector as well?

---

> ### Author Response · Authors · 2023-11-17
>
> Thanks for the detailed and constructive comments to help our work, acknowledging our approach ```highly meaningful```.
>
> Our response to each concern:
> 1. ***While SimPLR is able to get comparable/better performance with ViTDet, it is still behind sota object detector (e.g. DINO), it is not clear how SimPLR can be transferred to different advanced detectors.***
> - As mentioned above, we preserve a simple design of decoder and focus more on fixing hierarchical and multi-scale constraints that appear in backbone and encoder. Therefore, techniques like *object query denoising* in DINO or *hybrid-matching* in other works can still be applied in our detector to further improve the performance.
> - It can also be seen that without Object365 pre-training, DINO/MaskDINO achieves comparable performance with our SimPLR+ViT-H, and SimPLR+ViT-H is still faster than DINO/MaskDINO+Swin-L
> 2. ***The proposed method is based on a specific box-attention approach, can such method be applied to other transformer-based detector as well?***
> - In our work, we find that encoding scale information into the attention computation helps the model to learn scale-equivariant features. I think the same strategy can be easily applied to deformable attention as well. However, we choose box-attention because it shows better performance and it can do both detection and segmentation tasks.

---

### Author Response · Authors · 2023-11-17

We would like to thank all the reviewers for spending their time and providing valuable feedback to our work.

Overall, we'd like to highlight the contribution of the paper:
- Our goal is to pursue a plain detector whose backbone *and* detection head are both *single-scale* and *non-hierarchical*
- When we agree that our proposed method is simple, it is not straight-forward as commented by Reviewer SZaB. Instead, it is more difficult to find a simple but effective method:
    - Concurrent to our work, Lin et al. [a] introduces PlainDETR which also removes multi-scale input. However, it still relies on multi-scale feature maps to generate object proposals.
    - In the decoder, PlainDETR also uses hybrid matching [b] to strengthen its prediction, while our decoder preserves a simple design as in DeformableDETR or BoxeR.
    - Our plain detector outperforms PlainDETR by 1.6 AP point in plain object detection with ViT backbone
    - We show it works on segmentation tasks: instance segmentation and panoptic segmentation
- Our study shows that encoding scale information into the attention can learn the scale-equivariant features that removes hierarchical and multi-scale constraints
- We show a good scaling behaviour with a plain detector, suggesting the strong potential of this direction.

[a] Lin et al. DETR Doesn't Need Multi-Scale or Locality Design. In ICCV 2023.

[b] Jia et al. DETRs with Hybrid Matching. In CVPR 2023.

[c] Chen et al. You Only Look One-level Feature. In CVPR 2021.

While we believe our work provides a good insight in learning scale-equivariant features for object representations, we would like to withdraw our paper and submit it to future venue.